# On the Potential of Silicon as a Building Block for Life

**DOI:** 10.3390/life10060084

**Published:** 2020-06-10

**Authors:** Janusz Jurand Petkowski, William Bains, Sara Seager

**Affiliations:** 1Department of Earth, Atmospheric, and Planetary Sciences, Massachusetts Institute of Technology, 77 Mass. Ave., Cambridge, MA 02139, USA; bains@mit.edu (W.B.); seager@mit.edu (S.S.); 2Department of Physics, Massachusetts Institute of Technology, 77 Mass. Ave., Cambridge, MA 02139, USA; 3Department of Aeronautics and Astronautics, Massachusetts Institute of Technology, 77 Mass. Ave., Cambridge, MA 02139, USA

**Keywords:** silicon-based life, alternative biochemistry, alternative solvents, sulfuric acid biochemistry

## Abstract

Despite more than one hundred years of work on organosilicon chemistry, the basis for the plausibility of silicon-based life has never been systematically addressed nor objectively reviewed. We provide a comprehensive assessment of the possibility of silicon-based biochemistry, based on a review of what is known and what has been modeled, even including speculative work. We assess whether or not silicon chemistry meets the requirements for chemical diversity and reactivity as compared to carbon. To expand the possibility of plausible silicon biochemistry, we explore silicon’s chemical complexity in diverse solvents found in planetary environments, including water, cryosolvents, and sulfuric acid. In no environment is a life based primarily around silicon chemistry a plausible option. We find that in a water-rich environment silicon’s chemical capacity is highly limited due to ubiquitous silica formation; silicon can likely only be used as a rare and specialized heteroatom. Cryosolvents (e.g., liquid N_2_) provide extremely low solubility of all molecules, including organosilicons. Sulfuric acid, surprisingly, appears to be able to support a much larger diversity of organosilicon chemistry than water.

## 1. Introduction

Life in environments very different from Earth’s could have a biochemistry that is very distinct from Earth’s life [1,2,3,4,5,6]. A persistent contender as an alternative elemental basis for life is silicon, mainly due to the chemical analogies between silicon and carbon. The idea that life could be based on silicon rather than carbon was first proposed in 1891 by the German astrophysicist Julius Scheiner [7]. Silicon shares many similarities with carbon and is, in the form of various silicates in rocks, the second most abundant element (after oxygen) in Earth’s crust [8]. However, despite its abundance, terrestrial life only uses silicon as silicic acid and silica (see Appendix A). There are no known examples of life using any other type of silicon chemistry. When people speak of “silicon-based life”, they are usually referring to the diverse organosilicon chemistry, and specifically chemistry in which carbon in organic molecules is replaced by silicon. So why is life’s use of silicon so restricted, and could a wider silicon chemistry be used in other biochemistries?

The possibility of silicon-based life is usually either accepted as plausible (e.g., [1]) or dismissed out of hand (e.g., Carl Sagan famously called himself a “carbon chauvinist” [9]). However, despite a century of speculation and experimental work on organosilicon chemistry, the plausibility of silicon-based life has never been systematically addressed nor objectively reviewed. This paper seeks to fill this gap. We review what is known, modeled, or speculated about silicon as a basis of life, and provide a comprehensive assessment of the opportunities for silicon in a non-terrestrial biochemistry, with a detailed discussion of the arguments for and against silicon as a building block of biochemistry.

We start with the discussion of the general requirements for the chemistry of life, no matter its chemical basis (Section 2). We next provide the overview of silicon chemistry (Section 3.1), including discussion of the diversity of silicon chemistry (Section 3.2) and thermodynamics of formation of silicon compounds (Section 3.3). We then discuss how silicon chemistry meets the requirements for the chemistry of life, focusing on the classes of environments in which life using silicon as a component of biochemistry could occur (Section 4). We conclude first that in water or ammonia silicon cannot be the main building block for life but might be used as a rare heteroatom, second that cryosolvents are poor environments for any biochemistry, and third that in sulfuric acid silicon could be used much more widely by life (Section 5). We relegate discussion of the details of how terrestrial life uses silicon to Appendix A, and more detailed background on silicon chemistry and speculations on silicon biochemistry to Appendix B, Appendix C and Appendix D.

## 2. General Requirements for the Chemistry of Life

We begin by describing the general requirements that any biochemistry must fulfil. The general characteristics that have to be met by any chemical basis for life can be condensed to three essential requirements: sufficient chemical diversity (Section 2.1); stability and reactivity (Section 2.2); and the presence of a solvent (Section 2.3) (reviewed in [10]). Although chemical diversity, reactivity, and solvent requirements are linked, we discuss them separately below. If silicon chemistry is to be a component of any biochemistry, then it must fulfil these requirements.

### 2.1. Chemical Diversity

Life needs a diverse set of chemicals with different functions. On Earth, the diverse set of chemicals are amino acids (to make proteins), sugars, and nitrogenous bases (to make nucleic acids), hydroxyl and keto acids (as core metabolic intermediates), lipids (to make membranes), and more. To build a diverse set of chemicals, life needs a set of elements capable of building molecules composed of many atoms that will provide sufficient biological functionality.

Sufficient chemical diversity needed to build a molecular repertoire suitable for life can only be achieved by a scaffolding element bonded with heteroatom elements. The scaffold atom is one that can join in chains and clusters to construct the skeleton or shape of a molecule, and the heteroatoms provide chemical activity in a molecule. Scaffolds provide the ability to make large molecules, and hence a large number of different molecules (compare the number of stable molecules of the type X_n_H_m_ that can be formed with X=Nitrogen (three–NH_3_, N_2_H_4_, N_2_H_2_) and with X=Carbon (an essentially infinite number of hydrocarbons)). Scaffold elements and heteroatoms tend to be different atoms. The scaffold needs to be relatively stable and unreactive, while at the same time bonding to functional atoms (heteroatoms) that provide chemical functionality and distinctiveness to each molecule (Figure 1).

On Earth, carbon forms the major scaffold element (hence the term “carbon-based life”). Several other non-metal elements could be viable scaffold alternatives to carbon in that regard [1,2,3,4]. Sulfur, boron and, in particular, silicon, are capable of forming covalent compounds in which many atoms of the same type are bound together to form large molecules, and therefore could, in principle, be considered as “scaffolding elements” [10] (Figure 2). Sulfur-based polymers, e.g., amphiphilic polythionates [11], are, however, limited to linear chains, which severely restricts the diversity of possible shapes in sulfur-based systems. Boron, on the other hand, forms polymeric structures that are clusters of atoms rather than smaller, isolated molecules, e.g., decaborane (14), B_10_H_14_, which, while stoichiometrically a boron analogue of the hydrocarbon decane, is in structure more similar to a diamond nanoparticle [12]. Among these alternatives, silicon seems to be the most promising choice as a substituent for carbon in biochemistry (Figure 2).

Because the scaffolding of carbon (and hydrogen) alone provides only the very limited chemical function required for metabolism, heteroatoms are needed to provide chemical reactivity. Heteroatoms can form covalent bonds to carbon (or other scaffolding elements) but are more electropositive or electronegative. This difference in electronegativity provides the reactivity of molecules (see Section 2.2). Heteroatom reactivity is responsible for the great majority of the metabolic reactions occurring in Earth’s organisms [13]. Life on Earth uses several heteroatoms (e.g., O, N, S and P) that form covalent bonds with the carbon scaffold of biomolecules and with each other.

Heteroatoms can conveniently be divided into common heteroatoms and rare heteroatoms (although this is for convenience and does not represent any absolute class of abundance, which in any case will vary from organism to organism). Common heteroatoms (e.g., nitrogen or oxygen in terrestrial life) are used in many different molecules in biochemistry. On Earth, oxygen dominates among the heteroatoms used by life. In fact, the oxygen heteroatom is so ubiquitous that Earth biochemistry has been described as the “chemistry of carbonyl groups” [4]. By contrast, rare heteroatoms are used in a small number of specific contexts. Examples in terrestrial life include selenium and fluorine. It is plausible that these elements are difficult for life to use widely but confer specific advantage in the rare molecules in which they are found. For example, the thermodynamics of fluorine make it hard to incorporate into organic molecules, but the toxicity of the resulting molecules makes fluorine valuable in plant defense chemicals [14].

We note that the chemical diversity in small molecules implies a potential chemical diversity in polymers. Proteins are enormously diverse because they can be made from a chemically diverse set of amino acid monomers.

Together, scaffolds and heteroatoms provide the chemical diversity and reactivity required for life.

### 2.2. Chemical Stability and Reactivity

Chemical stability and reactivity are the second general requirement of the chemistry of life. There must be a balance between biochemicals’ stability and reactivity in their solvent. Biochemicals must be stable to reaction both with the solvent and with each other, over whatever timescale is required for their biological function. Thus, for example, sugars cannot be a component of a biochemistry that uses concentrated sulfuric acid as a solvent, because sugars are dehydrated to amorphous carbon in under a minute in sulfuric acid conditions [16]. However, biochemicals have to be reactive in their solvent to some extent, in order to be able to perform their required biological functions.

Aprotic solvents (those that cannot donate protons) tend to be less reactive than water, ammonia, and sulfuric acid. Furthermore, reactivity is highly reduced at low temperatures so almost any chemistry is stable in liquid methane or liquid nitrogen. We would therefore expect very stable chemicals not to be components of life in cold, aprotic solvents, as they would not be reactive enough.

Stability and reactivity are therefore a function of the solvent as well as the chemistry itself. We discuss the solvent requirement next.

### 2.3. Solvent

Chemical life has to operate in a medium that allows molecules to move (as do liquids and gases) but is dense enough to stop large molecules from simply falling out as aggregates (as are solids and liquids). This means that the chemistry of life has to operate in a dense fluid solvent, most plausibly a liquid. Water is widely considered the ideal, possibly the only, solvent for life [17]. Water actively assists in the self-organization of membranes and polymers of life, therefore assisting in compartmentalization, a phenomenon that is likely to be a general requirement for life [18]. On Earth, liquid water also plays an active role in life’s metabolic processes.

Although water is a good solvent for Earth’s carbon-based life, many other solvents have been proposed as well suited for alternative biochemistries (e.g., sulfuric acid, carbon dioxide, hydrogen cyanide, propane, ammonia, hydrogen sulfide, ethane and methane, nitrogen, and even neon and argon) [1,3,4,19,20,21,22]. These liquids broadly fall into the protic solvents (solvents that, like water, can donate or accept protons from chemicals dissolved in them, such as H_2_SO_4_, H_2_S, NH_3_), and aprotic solvents (such as liquid hydrocarbons or nitrogen). In general, protic solvents are chemically aggressive, and their aggressiveness limits the chemistry that can stably dissolve in them. In contrast to protic solvents, aprotic solvents are generally less reactive and stably dissolve a wider range of chemicals than protic solvents.

Solubility of solids in any solvent generally increases with temperature, so cosmically common aprotic solvents such as liquid methane and nitrogen are poor solvents because they are liquid only at very low temperatures. Thus, the nature and temperature of the solvent in which life operates affects both what scaffolds are viable and what heteroatom chemistry is stable in that solvent.

## 3. Overview of Silicon Chemistry

We start the assessment of silicon as a building block of biochemistry by presenting an overview of the basic physico-chemical properties of silicon chemistry compared to carbon (Section 3.1), followed by the discussion of the chemical diversity of silicon chemistry as compared to carbon (Section 3.2). We then address the thermodynamics of formation of silicon chemicals, with a comparison to carbon equivalents, and discuss the thermodynamic barriers in the synthesis of complex silicon molecules. We present new findings on the possible diversity of silicon chemistry based on thermodynamic limitations (Section 3.3).

### 3.1. Silicon Chemistry Overview in Comparison to Carbon

Silicon is the closest analogue of carbon. Both silicon and carbon are tetravalent atoms that form primarily covalent (non-ionic) compounds. However, there is surprisingly little similarity between them that goes beyond the statement that both elements “can form four covalent bonds”. In this section, we summarize the similarities and differences of silicon and carbon and discuss the consequences of their different reactivities and chemical properties on the formation of complex chemistry.

The covalent radius of a silicon atom is larger than that of carbon which results in generally longer bond lengths and different bond angles. Different bond lengths and angles have especially large effects in ring structures containing silicon atoms, resulting in distinct ring conformations and ring reactivity as compared to their carbon-containing counterparts (see for e.g., sila-venlafaxine (**9**) in Section 3.2.2 below).

The bonding preferences of silicon are also different than carbon, mainly due to the availability of low-lying 3d orbitals that allow silicon to form compounds that have five- or six-coordinated silicon atoms. Especially, penta-coordinated silicon compounds are readily formed, which allows for many more reaction paths between tetra-coordinate silicon compounds than are available to carbon [23,24]. Stable penta- and hexa-coordinate silicon compounds are “super-chiral”.

Silicon is more electropositive as compared to C, N, O, and H. The higher electropositivity of Si creates an electron-deficient center in silicon and results, e.g., in a stronger bond polarization as compared to analogous carbon bonds, or in a reversed bond polarization of the C–H and Si–H bonds (Table 1). As a result of those differences, most bonds that silicon forms with non-metals are more strongly polarized than their carbon counterparts and thus more susceptible to electrophilic and nucleophilic attack. Even bonds that are considered to be very stable, like Si−C, have higher reactivity as compared to their carbon analogues. For example, silicon tetrachloride (containing Si–Cl bonds) is hydrolyzed almost instantly in water, whereas carbon tetrachloride (containing analogous C–Cl bonds), which is also thermodynamically unstable to hydrolysis, is stable for years in the presence of water. Silanes (SiH_4_, Si_2_H_6_ etc.) are stable as pure chemicals for many years but are very sensitive to water in the presence of trace alkali, unlike alkanes. Reactions of both disilane (Si_2_H_6_) and ethane (C_2_H_6_) with oxygen are clearly exothermic, but ethane may be mixed with oxygen at 200 °C without reacting, whereas disilane spontaneously combusts in air at 0 °C [25]. We discuss the implications of such differences in reactivity in Section 3.2.2.

The differences between bond energies for carbon- and silicon-containing chemical bonds are also reflected by the disparities in the reactivity of silicon and carbon-containing compounds (Table 1). The strength of a C–C bond is a little bit greater than of a C–O bond, while the analogous Si–O bond is much stronger than the Si–Si bond [26], although the exact bond energy depends on the substituents on the C or Si. As a result of these differences in individual bond strengths, the chemistry of organic carbon molecules is dominated by C–C polymerization (catenation, the formation of long chains of covalently linked carbon atoms, e.g., in hydrocarbons). Even though silicon is capable of formation of Si–Si catenated structures (e.g., in silanes), they are much more reactive than their C–C counterparts (especially in water). As a consequence of the greater reactivity of the Si–Si bond, the most common stable polymers of silicon are built from Si–O chains, as the Si–O bond is disproportionately stronger than any other Si-containing bond (Table 1). Moreover, the polymerization of silicon often leads to a meshwork of Si–O chains and not linear polymers like for carbon—recall that the formation of long linear polymers is often cited as an absolute, general characteristic of any biochemistry [27]. As a result, Si chemistry in oxygen-rich environments (e.g., water) ultimately leads to silica (SiO_2_) (which is a refractory solid rather than a gas, with no double bonds to oxygen as in its carbon equivalent CO_2_) (Figure 3).

### 3.2. Diversity of Silicon Chemistry

In this section, we explore the potential diversity of silicon chemistry from three viewpoints: the theoretical diversity of organosilicon chemistry; the observed functional diversity of silicon molecules; and the potential diversity of silicon-based polymers. We conclude with a new assessment that there is, in theory, sufficient potential diversity in silicon chemistry to build biochemistry.

#### 3.2.1. Theoretical Diversity of Silicon Chemistry

Silicon can form stable covalent bonds with the same crucial elemental building blocks as carbon. It can covalently bind itself, carbon, nitrogen, oxygen, sulfur, phosphorus, and the halogens, as well as semi-metals like germanium. Silicon can also form covalent bonds with many metals [31,32]. On top of the versatile binding to many other elements in its most common tetra-coordinate state, Si can also form stable penta- and hexa-coordinate compounds with crucial biogenic elements like nitrogen, carbon or oxygen, with or without overall charge on the molecule (see Table 1 for detailed description of the physico-chemical properties of Si). Moreover, the coordination numbers of silicon (3, 4, 5, 6) are different than that of carbon (1, 2, 3, 4). Such differences often result in the formation of silicon chemicals that do not have direct carbon analogues, emphasizing that the potential chemical flexibility of silicon (although achieved through different means) could be as great as that of carbon.

There are two types of measure of chemical diversity: structural and functional. Hydrocarbons show a staggering structural diversity but are chemically monotonous. As noted in Section 2.1, there are only three stable N_m_H_n_ compounds, but they show more divergent chemical behavior than all the hydrocarbons. In this section, we address the structural diversity of organosilicons; in the following section (Section 3.2.2), we discuss the functional diversity of silicon.

To illustrate the scale of possible structural diversity in organosilicon chemistry as compared to carbon, we generated a set of possible chemical structures that are composed of up to seven atoms selected from S, P, O, N, F and Cl as heteroatoms, (with as many additional H atoms as was required to satisfy valence rules), using an algorithm derived from that described in the work of [33,34]. In short, the algorithm takes a set of topological structures defined as alkanes and substitutes heteroatoms into them according to bonding rules. Starting structures and bonding rules are defined to reflect stability and reactivity in different solvents (see Appendix E.1 for more details on the calculations of the silicon chemical space).

We generated sets of structures likely to be stable in water, sulfuric acid or in a hypothetical aprotic cryosolvent (we did not consider solubility in cryosolvents for the chemical diversity estimation; see Section 4.4 for discussion of the solubility limitation of cryosolvents), generated with or without silicon atoms. These calculations do not include penta- and hexacoordinate systems, which are under-represented in small molecules (by definition, there are no 5-atom molecules in which a silicon atom is bonded to five other atoms).

From our combinatorics study, we find that adding silicon to the repertoire of the chemistry possible in water adds little additional structural chemical diversity (Figure 4). Such a very modest increase in chemical diversity is due to the fact that almost all Si–X bonds, including Si–Si bonds, are readily attacked and hydrolyzed in water (Table 1). In contrast, we find that adding silicon to the repertoire of structures stable in sulfuric acid has a greater positive impact on the available chemical diversity. In aprotic solvents, where neither attack by H^+^ or OH^-^ is possible, the increase in chemical diversity is greater still. However, our calculations suggest that, overall, silicon does not have a major impact on the structural chemical diversity available to life (as compared to carbon). This is because carbon can readily form double and triple bonds with C, N, O and S, whereas silicon double bonds are strongly disfavored (although see Appendix C for an overview of exceptions to this generalization about silicon chemistry). Such limitations of silicon chemistry are further exemplified by the efforts to make silicon analogues of terrestrial biochemicals, which have only revealed how different silicon and carbon really are [35]. We discuss the observed functional diversity of silicon chemistry next.

We conclude with a few points. First, the vast potential theoretical space of silicon chemistry is almost entirely unstable in water, and hence not available to a biochemistry based on water as a solvent (see also Section 4.1). This major point has been implied from several examples (e.g., [1,3]) but never fully emphasized. Second, the presence of Si adds a greater diversity of molecules—perhaps enough to contribute as a heteroatom component of biochemistry—to the available chemistry of hypothetical life based in sulfuric acid (see also Section 4.3). Third, the majority of the chemical space of silicon chemistry is available in aprotic solvents such as dry liquid hydrocarbons or liquid nitrogen.

#### 3.2.2. Observed Functional Diversity of Silicon Chemistry

The functional diversity of silicon chemistry is demonstrated by the wide range of compounds that have been created in the lab, and by their very varied chemical properties, including even some biological uses. We can only provide a brief overview of the diversity of molecules containing silicon (either as a scaffolding element or as a major heteroatom) that are available to silicon chemistry. Such an overview will serve as an illustration of a proof of principle for the chemical diversity of synthetic organosilicon molecules. We discuss several such examples below.

Silanes (silicon analogues of hydrocarbons), both branched and unbranched [15], and diverse cyclosilanes (e.g., cyclohexasilane) are commonly known silicon analogues of hydrocarbons [36]. Polymeric siloxenes (cyclosilane rings with attached –OH groups) have no direct carbon analogue, but are reminiscent of functionalized graphenes [37]. The linked siloxene ring structures have structural and electronic similarities to the porphyrin units of heme and chlorophyll [37] (see Appendix C for the discussion of hypothetical, analogous solutions of silicon to the common biological functions). Many complex cage systems, composed of multiple connected ringed systems, are also known. One example is silsesquioxanes, containing Si–O-bond-rich “core” structures that can be modified with other groups that allow for a precise spatial orientation of the entire molecule, therefore opening the possibility for the complex regulation of the chemical accessibility of the core structure [38,39,40,41].

Mere structural diversity in general is insufficient to support a biochemistry (Section 3.2.1). The chemistry of life must support chemical functional diversity as well. Silicon provides the basis for diverse chemical function when combined with other atoms (other than carbon). Several examples include Si–S-bond-containing silicon analogues of thiols (thiosilanes) [42,43,44] and Si–N-bond-containing α-sila-amino acids [45]. Dynamic, highly polarized silicon systems, where rapid, reversible chemistry occurs, are also known [46]. Silicon can also provide electron conduction down an Si–Si silane chain [47,48].

Diverse silicon chemicals that have organic carbon scaffolds around the silicon atom (i.e., where silicon is acting as a heteroatom, not a scaffold element) are also known, although many react very rapidly with water. Examples of such chemicals include zwitterionic silicon compounds [49]; a range of organosilicon molecules with negatively charged silicon centers [50,51,52]; positively charged silicium (tricoordinate silicon [53]); and pentacoordinate silicons [54,55], some of which have silicon bonded to five different atoms at once [23], that can have useful catalytic properties in carbon–carbon bond formation [56].

Silicon can form stable structures inaccessible to carbon, such as stable gem diols, which have been shown to both be stable in water and to be pharmacologically promising [57] (see also Section 4.1). Silyl fluorides have also shown specific biological effects [58], as have other, more exotic, molecules like hexa-coordinate silicon (**10**) [59].

When silicon is incorporated into an organic compound, the chemical and physical differences contributed by the silyl group can provide such compounds with unique chemical properties, which can lead to subtle biological function different from carbon. These new unique properties lead to useful pharmacological and medicinal applications of organosilicon molecules and open new avenues for unique specific interactions between organosilicon molecules and biological macromolecules. Good examples of such new biologically and pharmacologically relevant properties were shown for sila-venlafaxine (**9**), where the introduction of silicon into the ring structure provides a conformation that is not accessible with the corresponding carbon analogue [58].

One of the persistent myths about silicon chemistry is that it is “monotonous”, and this is in part because many of the key analogous functional groups in carbon-based biochemistry cannot exist in silicon chemistry. Thus, despite decades of effort, no-one has succeeded in making a compound with a –H and an –OH group attached to the same silicon atom (Reinhold Tacke, per comm). Thus, direct silicon analogues of primary alcohols or sugars are impossible. Similarly, silicon doubly bonded to oxygen is not stable under Earth surface conditions even in the absence of water or molecular oxygen, so silicon analogues of aldehydes, ketones and carboxylic acids are impossible. However, this is not the point at issue. The question is whether silicon can provide diverse function, and, as we argue on the examples presented above, it can. Silicon does so, however, through different chemistry than carbon. We discuss the capability of silicon for complex, diverse chemical functionality (on the example of polymer chemistry) next.

#### 3.2.3. Observed Structural Diversity of Complex Silicon Polymer Chemistry

As a subset of observed silicon chemical diversity, we turn to silicon polymer chemistry. Functional chemical diversity provided by silicon is only useful to life if silicon can form polymers. Virtually every biological process uses polymers. While the specific long polymers used by life on Earth, like proteins, DNA or RNA, do not have to be universal for all life, it is likely that the utilization of some form of polymers is one of the required general characteristics of all life, no matter its chemical composition [2,13,27]. It is therefore prudent to ask if silicon can provide sufficient chemical diversity, beyond small molecules, as a building block of complex polymers for biochemistry.

The capability of silicon polymer chemistry to form sufficiently diverse molecules to support complex biochemistry is often called into question. Silicon chemistry is often called “monotonous” in comparison to possible carbon equivalents [60]. The myth of the chemical monotony of organo-silicon polymers and small molecules likely stems from the comparison of the enormous diversity of carbon-based molecules produced by life on Earth with the known organo-silicon polymers used in industry. Clearly, most industrial organo-silicon polymers are monotonous, but so are most industrial carbon polymers, especially those that are not inspired by natural products. Chemical monotony is often required by the intended function of the industrial polymer. Chemical monotony is especially prevalent in plastics, but it is not an inherent feature of organosilicon chemistry. There is no inherent chemical reason why diverse high-molecular-weight silanes or siloxanes could not have a highly diverse and highly structured set of side chains.

In fact, the modern copolymer industry has many examples of very diverse organosilicon polymers, containing many different silicon-bearing monomers in single- or cross-linked polymeric chains [61,62,63,64]. (Many industrial polymeric structures are produced on a very large scale, e.g., popular silicones are produced in an impressive amount of two million metric tonnes per year [65])). Silicon is known to form stable polysilane polymers with an –Si–Si– scaffolding, containing as many as 500 consecutive Si–Si bonds [66]. Shorter chiral oligosilanes, with up to 26 consecutive Si–Si bonds, were also synthetized [2]. Such oligosilanes are capable of supporting many different side-chain groups of variable chemical functionality, including carboxylic acid groups (**1**) that are soluble in water and can self-aggregate into amphiphilic vesicles and micelle like structures or alkyl side chains that are soluble in non-polar solvents (**2**) [2]. Note that carbon often plays the role of a heteroatom in polymers that are scaffolded by silicon.

There is, therefore, little doubt that the true chemical diversity of silicon-bearing polymers is sufficient to build a complex scaffolding of biochemicals, analogous in their biological functionality to proteins, nucleic acids or carbohydrates in Earth’s biochemistry.

The capability of a scaffolding element to build diverse molecules, especially polymeric structures, is likely one of the universal requirements for life, no matter its underlying chemical composition. Silicon clearly meets that requirement. Furthermore, silicon chemistry can also lead to the formation of molecular barriers necessary to achieve compartmentalization, another likely universal requirement for life. For example, formation of complex macromolecular assemblies that can potentially serve as molecular barriers akin to lipid bilayers in Earth’s carbon biochemistry is also known. Examples of such large flexible systems include sheets, tubes, strings and many other shapes that can be formed by various derivatives of silane polyols [67,68,69]. Even on Earth, life recognizes a unique advantage of silicon as a useful structural element (see Appendix A).

The substitution of Si for C in the context of the Si–C bond has been successfully demonstrated for a series of pharmaceutically and biologically important compounds, including amino acids (e.g., (**3–6**)) [70]. In some cases, the silicon-containing group is incorporated as a side chain substituent, in peptide-like backbone-containing molecules (e.g., in silanediols). Only very recently, the silicon atom been incorporated at the central α-position of an amino acid (**7**, **8**) in a peptide chain forming short silicon-based peptides such as (**7**) [45]. Synthesis of α-sila-peptides (**7**) further exemplifies the potential for silicon as a heteroatom in the context of aqueous biochemistry (see Section 4.1).

We conclude that silicon chemistry can provide equivalent diversity of function to carbon chemistry, both from the point of view of theoretical as well as synthetic functional chemical space.


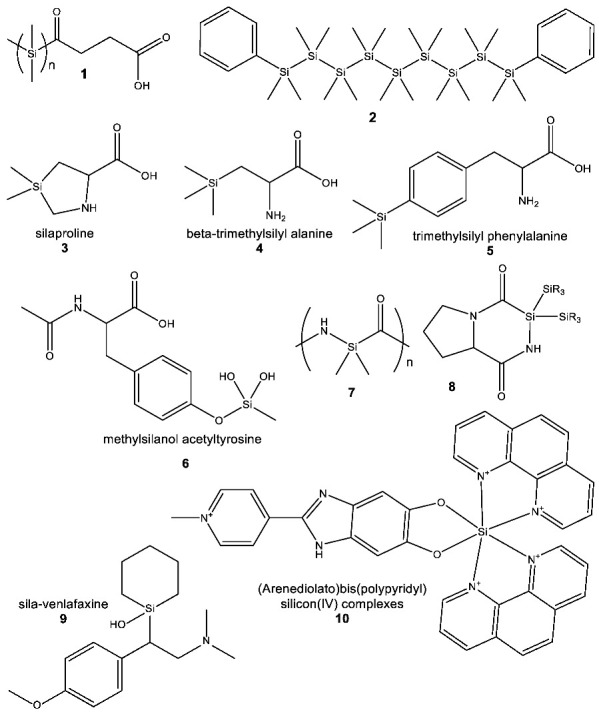


### 3.3. Thermodynamic Aspects of Formation of Silicon Compounds

We now address the thermodynamics of formation of silicon chemicals, providing a new analysis of energies of formation of silicon chemicals (see Appendix E.2 and Table 2). While thermodynamics is never an absolute barrier to life, chemicals that are energetically costly to make might be underrepresented in metabolism if less energetically costly functional equivalents are available.

In this section, we discuss whether there is a thermodynamic barrier to the formation of a diverse space of silicon-containing molecules and explore if basic thermodynamics could be the reason for the improbability of silicon-based biochemistry.

All compounds containing silicon bonded to atoms of similar electronegativity (Si, P) require substantially more energy to make than equivalent carbon compounds (Table 2). However, silicon bonded to atoms of greater electronegativity (including C) is relatively more thermodynamically favorable (negative ΔG of formation from SiO_2_) (Table 2). Note that SiF_4_ is actually more thermodynamically stable than CF_4_. This observation is consistent with the fact that silicon tetrafluoride is stable and produced as a trace gas by volcanoes on Earth [71].

Our thermodynamic calculations show that the energies of formation of Si-containing compounds are generally much higher, and therefore less favorable, than their carbon counterparts (Table 2). Such a high thermodynamic cost of the synthesis of silicon compounds could contribute to the scarcity of silicon in biochemistry; it is not, however, an absolute limitation.

## 4. Potential for Silicon Biochemistry in Different Solvents

A solvent provides the immediate chemical environment for life. In this section, we discuss the general requirements for Si biochemistry through the lens of the solvent, by assessing the degree to which several of the cosmically abundant liquids can support complex silicon chemistry. In this paper, we focus on the examples of the most plausible and cosmically abundant solvents; we note, however, that some speculations have suggested the possibility of liquid rock as a solvent for life [6,78]. Feinberg and Shapiro postulate a planet “Thermia” with a surface temperature above the melting point of rock and silicate-based life “swimming” in it [78]. Although such a liquid would be composed of silicates, it is not a plausible host for silicon-using biochemistry. Covalent bonds in silicate rocks are Si–O bonds, the strongest bonds silicon forms (Table 1). All the covalent bonds in solid rock are mobilized on melting, becoming no more stable than the hydrogen bonds in liquid water. Of necessity, no complex silicon-based compounds could be stable in an environment where the silicon–oxygen bonds are being broken on a millisecond timescale by heat (although we note that planet Corot-7b could be a real-Universe model for Feinberg’s and Shapiro’s planet “Thermia” [79]).

Throughout, when we discuss the abundance or rarity of silicon in biochemicals, we refer to the number of chemical structures containing silicon, not the total mass of silicon compounds in the biosphere.

### 4.1. Silicon Biochemistry in Water

Only a tiny fraction of the theoretical chemical space of silicon chemistry can be stable in water (Section 3.2.1). In fact, some of the commonly held views about the low diversity of silicon chemistry come directly from the instability of silicon chemistry in water. Silicon chemistry in water also requires substantially more energy to access than equivalent carbon chemistry (Section 3.3). For all of the above reasons, we argue in this subsection that silicon is unlikely to be a scaffold element or a common heteroatom element in water. Silicon may be a rare heteroatom, found in a small number of chemicals where the stability and/or thermodynamic barriers are sufficiently minor.

The observation that the biochemistry of silicon in terrestrial organisms is extremely chemically limited is consistent with the limitations of silicon chemistry in water. All biological silicon-containing structures are derivatives of only one molecule (silicic acid, H_4_SiO_4_) (**11**) and its dehydration product, silica. In all of Earth’s life, the silicon atom is bonded exclusively to oxygen, forming a Si–O single bond. No naturally occurring lifeforms synthesize bonds between silicon and any other atom (see Appendix A for a detailed overview of silicon biochemistry in life on Earth).



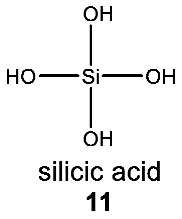



We now turn to discuss two scenarios in which Si chemistry can potentially be used in water-rich environments.


Silicon cannot be used as scaffold or major heteroatom element in water


Biochemistry based purely on Si–Si scaffolding is almost certainly impossible in water. Many Si–Si compounds hydrolyze almost instantaneously in water. The hydrolytic instability of the majority of silicon compounds is likely an unavoidable barrier to exclusive Si–Si scaffolding of life in aqueous environments [60] (Figure 3). Similarly, silicon as a major scaffold bonded to H, N, S or P atoms is implausible for its hydrolytic instability.

It is, however, possible to envision a scenario in which both carbon and silicon together play a role in scaffolding of biochemistry. The possible scaffolding scenarios for such Si–C “hybrid biochemistry” could include silanolate –Si–O–C– functional groups (not unlike the proposed cross-links in plant cell wall or vertebrate extracellular matrix, see Appendix A), although these are also hydrolytically labile. A hybrid backbone could also be built of silicon–carbon single bonds. The silicon–carbon single bond is a strong, water- and other-solvent-stable covalent bond that in principle could be utilized in biochemistry as a scaffolding bond (Table 1). Such use of silicon would be analogous to the role nitrogen plays in the backbone of proteins, or that phosphorus and oxygen play in the backbone of nucleic acids. However, including silicon as a major scaffolding element would have to give a very significant evolutionary advantage that would offset the tremendous energetic cost of mobilizing large amounts of Si.

In a water-rich environment, on a typical rocky planet where the C/O ratio is heavily skewed towards O, the main form of silicon will be sequestered in highly unreactive and insoluble silica-rich rocks (see Appendix B for the discussion of rare exceptions from this rule). The excess cosmic abundance of elemental oxygen, as compared to other elements that silicon could be stably bonded with, ensures that the great majority of the available silicon is almost exclusively bonded to oxygen. Similarly, a very high affinity of oxygen to silicon makes it unlikely that bonds between silicon and other elements (like Si–Si or Si–C) would be anything but a rare oddity in environments where oxygen is plentiful. It is, therefore, very likely that silicon chemistry on the planetary bodies in the Galaxy is dominated by the chemistry of silicon and oxygen in the silica rocks.

Utilization of silicon for building a rich and chemically diverse biochemistry in water necessities prior breaking of the Si–O bond, a feat that, as of yet, life on Earth appears to be incapable of doing. The incredible stability and strength of the Si–O bond makes hybrid Si–C scaffolding using Si as a major heteroatom element, tremendously energetically costly and therefore very unlikely. It does not mean, however, that such Si–C scaffolds for complex biochemistry are completely impossible. In environments on planets which have C/O ratios favoring C over O, or with much less overall O content (the hypothetical carbon planets [80]), the main building blocks for Si–C hybrid biochemistry might be more readily available (see Appendix D). It is important to stress, however, that such environments might be very rare.


Silicon can be used as a rare heteroatom element in water


We now present several examples of how Si could in principle be used as a rare heteroatom in water-rich terrestrial biochemistry.

The fact that life on Earth does not use silicon in any other capacity than silicic acid and silica is not in itself an evidence for an inherent limitation of life’s biochemical machinery. For example, life cannot make Si–Cl bonds in water. Similarly, Si–H bonds hydrolyze to Si–OH in a matter of a few hours under mammalian physiological conditions (e.g., [81]). But Si–C bonds that are stable in water could in principle be used by life (Section 3.1). No natural enzyme system can break Si–O bonds and synthesize Si–C bonds. Naturally occurring terrestrial enzymes process silicon-containing drugs and synthetic molecules via the carbon moieties, with silicon-containing functional groups left intact or readily hydrolyzed [58,82]. However, the possibility for silicon’s incorporation as a rare heteroatom in organic molecules by water-based life appears much more likely than previously thought, thanks to a series of elegant experiments in directed evolution by Frances Arnold’s laboratory. The experiments suggest that, at least in some capacity, life is capable of evolving means to create Si–C bonds previously not known to biology [83,84]. A chiral Si–C bond formation is catalyzed by an artificially evolved variant of *Rhodothermus marinus* cytochrome *c*. However, the formation of the Si–C bond happens in contrived conditions, in a thermodynamically favorable conversion of the Si–H bond in phenyldimethylsilane substrate to an additional fourth Si–C bond (Figure 5). Such experiments show that terrestrial biochemistry could generate Si–C bonds, but not how it could reduce silica to a silane as the feedstock for such a reaction.

The concept that terrestrial life could use silicon bonded to carbon as a rare heteroatom is supported by the example of boron in biochemistry. Boron (B) is present in Earth’s crust as borate, which, like silicate, requires substantial energy to reduce. Boron, like silicon, is used widely by life as an oxyacid. Compounds containing B–C and B–N bonds, like silicon compounds, are known to organic chemists and utilized in pharmacology, so there is clear evidence that organoboron compounds have useful biological functions [85,86,87]. However, unlike the case of silicon, there is a reported example of boron as a heteroatom in a natural compound containing a B–C bond (**12**) [88], suggesting that there is an evolutionary benefit to breaking the difficult B–O bond for inclusion of boron in biochemicals.



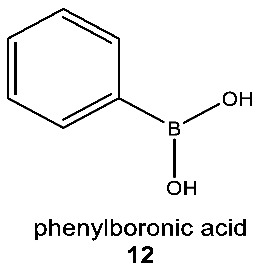



The example of a phenylboronic acid (**12**), a natural B–C-bond-containing compound isolated from cranberry fruit (*Vaccinium* sp.), shows that life is capable of overcoming the very high energetic cost of breaking and chemical transformation of the very stable B–O bonds if there is a useful function for such chemistry that cannot be achieved through other means (it is not known what that advantage is). It is very much possible that, similarly to organoboron, some small number of organosilicon natural products with silicon heteroatoms await discovery. Therefore, looking for analogous natural products containing a Si–C bond is not without merit, especially since, as we discuss below, organosilicon chemicals could provide life with a unique and useful biological function. Such an evolutionary advantage might be enough to offset the energetic costs of breaking the very stable Si–O bonds.

What might that selective advantage be? Clearly, it must derive from the unique chemistry of silicon, and one such chemistry is the unique chemical functionality of silanols, silanediols and silatriols (**13**, **14**, **15**).



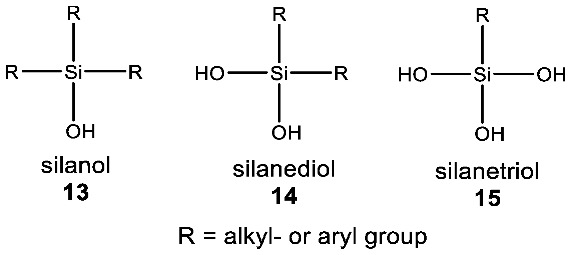



Silanols, silanediols, and silatriols (**13**, **14**, **15**) are silicon-containing analogues of alcohols that are characterized by unusual solubility properties, often being similarly soluble both in water and other solvents, like hexane [89,90,91,92]. Such solubility behavior is likely the result of the formation of strong hydrogen-bonded molecular complexes in solution [69,93,94]. Such enhanced hydrogen-bonding abilities and increased acidity of silanols, relative to carbon analogues, have potentially useful biological applications that their carbon analogues cannot provide (Figure 6). For example, silanediols are present almost exclusively as the geminal diol tautomer over silanone (i.e., Si=O). Silanediols mimic the geminal diol form of carbonyl as a transition state analogue in the catalytic cycle of proteases; they are therefore considered potent protease inhibitors [57,95,96,97,98,99]. This characteristic of silicon organic chemistry could be potentially explored for a useful biological function by water-based terrestrial life.

In conclusion, the usage of Si in the capacity of a rare heteroatom for water-based biochemistry can, in principle, provide life with unique biological functionality of silicon chemistry that cannot be provided by other heteroatoms. In theory, this advantage might counterbalance the significant energetic expense to an organism to mobilize silicon from inorganic silica. The water stability of Si–C and Si–O bonds makes them potentially attractive as carriers of useful biological functions and opens up the possibility for specialized utilization of Si as a rare heteroatom.

### 4.2. Silicon Biochemistry in Non-Aqueous Acid/Base Solvents

A number of protic solvents have some chemical similarity to water and could in principle be solvents for life. However, bearing chemical similarities to water, protic solvents pose challenges for silicon biochemistry.

The principle protic solvent similar to water is ammonia. Both water and ammonia are acid/base solvents that self-ionize to form a significant concentration of conjugate acid and conjugate base (H_3_O^+^/OH^−^ in water, NH_4_^+^/NH_2_^−^ in ammonia). The conjugate acid can act as an electrophile or Lewis acid, the conjugate base as a nucleophile or Lewis base.

Ammonia on planetary bodies is unlikely to exist as a pure solvent, for two reasons. First, ammonia vapor is easily photolyzed to produce molecular nitrogen, a process that is effectively irreversible outside of the deep atmospheres of giant planets. A substantial planetary ammonia ocean could therefore only be maintained if the planet atmosphere and surface is protected from UV radiation. Secondly, oxygen is cosmically more abundant than nitrogen, such that any environment with condensed ammonia would also have condensed water. Because ammonia and water are fully miscible in each other, the result would be a mixed water–ammonia ocean.

In any event, an ammonia solution is quite basic, and as a result very aggressive to silicon chemistry. As a strongly protonating solvent, we would expect ammonolysis (an analogous process to hydrolysis in water) to pose a serious limitation to any complex silicon chemistry in NH_3_ solvent. As far as we are aware, however, Si compound chemistry has not been studied in liquid ammonia.

Other protic solvents could include H_2_S and HCN, but they are not cosmically abundant and not expected to be commonly present on planetary bodies.

### 4.3. Silicon Biochemistry in Sulfuric Acid

Sulfuric acid is considered an even more chemically aggressive solvent than water, and, as a consequence, an implausible solvent for biochemistry. Terrestrial biochemistry is rapidly destroyed by concentrated sulfuric acid. However, we have found, unexpectedly, that a significant fraction of silicon chemistry is stable in the harsh conditions of concentrated H_2_SO_4_ (see Section 3.2.1; (Figure 4), despite sulfuric acid’s highly aggressive, polar, and chaotropic character. Below, we expand on the chemistry of silicon in sulfuric acid and assess the viability of silicon biochemistry in this unforgiving solvent.


Silicon chemistry is more stable in sulfuric acid than in water


Perhaps surprisingly, a larger fraction of silicon chemical space is stable in concentrated sulfuric acid than in water. This is because much of the instability of silicon compounds in water arises from nucleophilic attack by OH^-^ ions on the positive silicon atom (Section 3.2.1) and the stability of the resulting pentacoordinate structure. In contrast, in concentrated sulfuric acid, electrophilic attack dominates and silicon atoms, being electron-poor in almost all compounds, are not efficient targets of electrophilic chemistry.

Such a difference in reactivity means that a range of chemical groups are stable in sulfuric acid but relatively unstable in water. For example, trifluoralkyl groups (alkyl-SiF_3_) are almost instantly hydrolyzed in water, whereas they are stable in concentrated sulfuric acid [100]. Chlorosilanes take hours to days to hydrolyze in 100% sulfuric acid at 20 °C [101], whereas in water they hydrolyze effectively instantly. Alkyl silanes are resistant to cleavage of the Si–C bond under sulfuric acid conditions that will sulfonate an aromatic group [102]. Si–Si bonds are more stable in 100% sulfuric acid than Si–C bonds (that are, themselves, also stable in sulfuric acid if the carbon is aliphatic, not aromatic) [103]. Si–OH groups can be sulfated to form sulfate esters, depending on conditions [102]. Silane moieties (Si–H bonds) are stable to reaction with sulfuric acid at room temperature in some contexts—in others, where ring strain is present (e.g., silacyclopentane), they are hydrolyzed [104]. Note that Si–H groups are stable in pure water, but the slightest trace of alkali compounds, including the presence of ordinary glass, catalyzes their rapid hydrolysis [25]. There is no data available on the stability of Si-S and Si-N bonds in sulfuric acid.

There are very few exceptions to the above, Si chemicals that are more stable in water than in sulfuric acid. One notable example is that Si-phenyl bonds are readily broken in sulfuric acid, but not in water [105,106]. In addition, low-molecular-weight silicones are generally more readily rearranged into silanols or sulfate esters in sulfuric acid [107] than in water.


Possible advantages of silicon chemistry for hypothetical sulfuric-acid-based life


The fact that a larger number of silicon functional groups appear to be stable in concentrated sulfuric acid than in water opens the possibility for hypothetical life in sulfuric acid to use silicon to a considerable extent (Figure 4). We therefore turn to specific examples of chemical functionalities of silicon chemistry in sulfuric acid.

The first functionality comes from the very stable hydrogen bonds that silicon compounds can form in general. For example, Si–OH and Si–F bonds are highly polarized and would be expected to form extremely strong hydrogen bond donors and acceptors, respectively. The silicon–hydrogen bond strength could be valuable to overcome the chaotropic effects of sulfuric acid in forming stable macromolecular structures. The hydrogen bond energies between Si–F and H–X are not known, but the energy of the similar, very stable, H–F:H–OH dimer hydrogen bond is ~45 kJ/mol, compared to the HO–H:H–OH dimer of 21 kJ/mol [108]. With silicon being slightly more electropositive than hydrogen, H-bonds involving Si–F are expected to be even more stable than H–F ones.

Silanes (silicon molecules containing Si–Si bonds) can exclusively provide another potentially useful biological functionality for hypothetical sulfuric-acid-based life. The Si–Si chains, many of which are known to be stable in concentrated sulfuric acid, have a degree of σ orbital overlap that allows electron conduction down the scaffold of the molecule [47]. Such electron conduction is analogous to conjugated alkene systems in Earth life’s biochemistry. Conjugated alkenes such as isoprene are very rapidly attacked in concentrated H_2_SO_4_ [109,110], and so, in principle, long-chain silanes in sulfuric acid could substitute for biochemical functions carried out by conjugated dienes in terrestrial chemistry.

We also note that some silicon-containing polymers are highly resistant to sulfuric acid (e.g., polymers where silicon and carbon atoms alternate in the backbone, rather than the silicon–oxygen alternation of silicones, are stable to 98% H_2_SO_4_ at 90 °C [111]. Such unique silicon chemistry might also provide necessary biological functionality that is otherwise difficult to attain in sulfuric acid through exclusively carbon-based chemistry.

We emphasize that these examples of potential biological uses of silicon chemistry are speculations, not predictions. We use them here solely to illustrate that silicon has specific, potential advantages as a heteroatom for compounds in a sulfuric acid solvent—advantages that either do not apply, or apply less, in water. Adding silicon to the repertoire of structures stable in sulfuric acid has a greater positive impact on the available structural and functional chemical diversity than in water (Section 3.2.1 and Figure 4). This greater scope of stable silicon functional groups could result in greater evolutionary advantage for sulfuric-acid-based life (as compared to Earth’s water-based life) to use silicon chemistry. Not only is the size of available silicon chemical space in sulfuric acid greater than that in water, but the overrepresentation of stable silicon functional groups could offset the smaller number of carbon-based functional groups that are stable in the aggressive conditions of concentrated sulfuric acid.

Therefore, the evolutionary pressure for any sulfuric-acid-based life to explore silicon does not come solely from the advantages of the larger scope of available silicon chemistry but also from the potential necessity to explore silicon chemical space in sulfuric acid to perform biological functions.


Planetary environments with sulfuric acid


Sulfuric acid has been suggested to be an abundant solvent on the surface of planets [21]. However, unlike other speculated high-temperature solvents such as HCN and NH_3_, there is a precedent in the Solar System for the planetary-scale existence of liquid with concentrated sulfuric acid, and that is the Venusian clouds.

Venus has a temperate cloud layer (a region spanning from 48 to 60 km altitude with temperatures < 100 °C and pressures < 2 bar) believed to be composed of liquid sulfuric acid droplets. A permanent Venusian aerial biosphere has been a topic of scientific speculation for many decades (see, e.g., [112,113,114]). It is unknown what biochemistry could exist in such a highly reactive and aggressive protic solvent as sulfuric acid, but, as our discussion above indicates, a biochemistry that makes wider use of silicon is a possibility.

### 4.4. Silicon Biochemistry in Cold Aprotic Solvents

Silicon chemistry—really, any chemistry—is much more stable in aprotic solvents than in protic solvents. Aprotic solvents are non-ionizing solvents, which, unlike protic solvents (like water or ammonia), do not contain labile H^+^. In planetary terms, common aprotic solvents such as methane, ethane and nitrogen, that only form liquid phases at very low temperatures, are commonly called cryosolvents. Here, we use the term cryosolvent (short for cryogenic solvent, a solvent that is liquid at temperatures below −100 °C) for cold aprotic solvents.


Planetary environments with cryosolvents


Surface seas of aprotic cryosolvents might be a common occurrence on planets. In fact, aprotic cryosolvents like methane, ethane (C_2_H_6_) or liquid nitrogen may be the most abundant liquids on planetary surfaces, based on an exhaustive analysis of the propensity of stable surface oceans composed of liquids different than water [21]. N_2_ itself is a very common chemical on the cosmic scale, with abundances rivalling that of water.

Surface non-protonating solvents like liquid nitrogen (N_2_) oceans could be especially common on planets (or moons) orbiting M-dwarf stars [21]. The very low melting (−210 °C) and boiling points (−196 °C) of N_2_ necessitate that planets and moons hosting liquid N_2_ oceans have to receive far less incident stellar energy than planets hosting water oceans. The corresponding large planet–star separation (e.g., >1 a.u. for an M5 star [21] could mitigate the detrimental effects of high stellar activity of M-dwarf stars—planets orbiting close to the stars may be subjected to the catastrophic loss of an atmosphere from stellar flares. Such advantages could result in stable, “clement” conditions that could potentially allow for liquid N_2_ oceans to persist on a planetary surface for billions of years despite the relatively narrow temperature range for liquid N_2_ (−210 °C to −196 °C, at 1 bar). Such conditions could also exist in our own Solar System, on Neptune’s moon Triton, which orbits in an “N_2_ habitable zone” [115].

The low temperatures of aprotic cryosolvent seas pose at least two serious limitations as solvents for life (regardless of if such life is silicon- or alternative-carbon-based). The first problem is the low rate of any chemistry at such low temperatures. The second problem stems from the low solubility of molecules at cryogenic temperatures. Of those two limitations, the first, i.e., slow rates of chemical reactions, is easier to overcome.


Low chemical reactivity in cryosolvents


Slow reaction rates could be prohibitive for the formation of, or reactivity between, complex molecules. Chemical reactions occurring in cryosolvents would proceed very slowly, much more slowly than in Earth’s surface environment. The speed of chemical reactions generally drops by a factor of 2–3 for every 10 °C temperature decrease [116]. This drop of chemical reaction rate, however, is not an absolute limitation; it could actually be an advantage specific to silicon chemistry. The key factor in the formation of complex chemicals at any temperature is the selection of chemical reactions that are specifically tailored to a given temperature range [3]. Many silicon chemicals that are too reactive at Earth surface temperatures may have chemical reactivities “just right” at the temperature ranges of cryosolvents (including very low temperatures of liquid N_2_). (Silicon can do very fast chemical reactions at extremely low temperatures of liquid O_2_, as exemplified by experiments on the reactivity of amorphous silicon and oxygen [117].)

Specifically, two features of silicon chemistry support the notion that the reactivity of complex silicon chemistry could be uniquely suited for cryosolvent temperatures.

First, the increased reactivity of silicon—a disadvantage in water—could be an advantage in cryosolvents. Silicon is more electropositive than carbon, most Si bonds with non-metals are more polarized than the equivalent C bonds. As a result, such bonds are more liable to electrophilic and nucleophilic attack (see Section 3), allowing chemistry using weaker nucleophiles or electrophiles. Such reactivity is predominant in solvents like water (and ammonia). Such differences in reactivity are important because strong nucleophiles or electrophiles are themselves likely to be polarized and hence insoluble in cryosolvents (see below for further discussion on solubility). Additionally, common Si–X bonds are generally weaker than equivalent C–X bonds, and, as such, require less thermal energy to break for any given reaction mechanism. Again, weaker bonds might be considered an advantage over “classical” carbon chemistry in very-low-temperature environments.

The inherently greater reactivity of organosilicon-based chemicals could also be an advantage through enabling greater control and regulation of silicon-based biochemical processes. For example, chemical reactions involved in the formation and breakage of hydrogen bonds are much slower at cryotemperatures, which could allow for complex regulation of their formation by catalysts [4]. The much stronger nature of hydrogen bonds in cryosolvents could stabilize molecules to a much greater degree than in liquid water at Earth’s surface temperatures. While, sometimes, such stabilization effects might be viewed as a detriment (e.g., much stronger Si–OH H bonding), they could be beneficial for easy catalytic control of reactivity and stabilization of the genetic polymer molecules of hypothetical silicon-based life forms.

Secondly, in non-polar cryosolvents, many silicon-bearing functional groups that are completely unstable in water (including exotic unsaturated silanes) are stable and could in principle be utilized for useful biological function. Such higher structural and functional diversity of silicon chemistry in cryosolvents could make utilization of silicon chemistry much more attractive for life and could potentially offset the high energy requirements needed for cleavage of the Si–O bond and the mobilization of silicon from silica rocks. For example, functional groups containing multiple bonds between silicon atoms (e.g., Si=Si, Si=C, and Si#Si) are well known (see Appendix C, for detailed discussion of this unusual silicon chemistry), but so far are only known to be stable in sterically constrained compounds (i.e., compounds where the other silicon valences are occupied by very bulky groups) and in the absence of water, ammonia and a range of other groups, like carbonyls and alkynes [118,119]. In colder environments, such systems, though still reactive, are stable enough that their reactivity is much easier to regulate and control and, hence, much more useful.

Thus, the generally lower reactivity of chemicals at cryogenic temperatures is likely not a major barrier for silicon chemistry in cryosolvents.


Low solubility of chemicals in cryosolvents


The second, much more serious barrier for the possibility of complex organic chemistry of any kind in cryosolvents is the very low solubility of molecules (especially large complex polymers) under low-temperature conditions. The low solubility of molecules likely means that no cryosolvents are suitable for life. (There does not appear to be a solubility barrier in warmer solvents like water or sulfuric acid.)

Cryosolvents can in principle dissolve non-polar solutes. The solubility of non-polar molecules that do not form strong hydrogen bonds depends on their molecular weight as well as weak electrostatic interactions. However, due to the very low temperatures, even small non-polar molecules such as butane have very low solubility in liquid methane or liquid nitrogen. Polar molecules such as water, which form strong hydrogen bonds, are effectively completely insoluble in cryosolvents (e.g., [120]).

To assess the degree of the solubility limitation on the possibility of silicon biochemistry we estimate the solubility of silicon molecules at cryogenic temperatures. For our calculations, we use the example of liquid nitrogen (N_2_). We find that even the simplest silicon compounds have very low solubility in liquid nitrogen, confirming that the low solubility is likely the main limitation for life at cryogenic temperatures (Figure 7). We discuss the details of our calculations below.

We estimate the solubility of the silicon compounds listed in Table 2 in liquid nitrogen at −196 °C (its boiling point at 1 bar), using the modified linear free-energy relationship method of Abraham [121] (see Appendix E.3 for more details).

Our results illustrate that even the simplest silicon compounds—SiH_4_ and Si_2_H_6_—are expected to only have parts-per-thousand solubility in liquid nitrogen, and more complex molecules of complexity equivalent to amino acid glycine will have sub-parts-per-million solubility. The exception is silicon tetrafluoride, which is estimated to be anomalously soluble. For context, of the few thousand chemicals in Earth’s life core metabolism [143], probably only a dozen or so are soluble in liquid N_2_ at >ppm level; none of these have an –OH group or a molecular weight over 100. For comparison, the solubility of biochemicals in water is much higher, with solubilities reaching molar concentrations. Sugars alone, with the general formula CH_2_O, and any number of carbons from 3 (trioses such as glyceraldehyde) through 6 (hexoses such as glucose) up to 9 (such as sialic acid) are all soluble in water at molar concentrations, and can build thousands of possible water-soluble structures.

Life requires a diverse set of chemicals and a solvent, as summarized at the start of this paper. If a solvent cannot, even in principle, dissolve a diverse set of chemicals, then that solvent cannot support life.

We conclude that, despite the favorable conditions for stability and reactivity, the solubility barrier is detrimental for Si and any other chemistry in cryosolvents.

We summarize the interconnected nature of solubility and chemical stability in Figure 8. Water does indeed occupy an optimal position in this diagram, balancing reactivity with solubility.

## 5. Synthesis and Conclusions

We have reviewed silicon chemistry in comparison to carbon chemistry and highlighted key points related to the possibility of Si as a building block of life.

Silicon and carbon are “false twins”. Their similarities are superficial and insufficient to mitigate their crucial differences. Chemistry that is stable and normal for carbon is unstable and exotic for silicon, and, similarly, chemistry that is unstable and impossible for carbon is stable and routine for silicon. Silicon’s distinct chemical characteristics and reactivity make it a challenging choice for life (Section 3.1).Silicon-based life that uses Si exclusively as a scaffold element is often portrayed in science fiction. An iconic example of a fictional silicon-based life form is Horta from the twenty-fifth episode—“The Devil in the Dark”—of the first season of the popular science fiction television series Star Trek TOS, itself possibly based on the “Siliconey” in Asimov’s short story “The Talking Stone” (1955). However, silicon-based life that uses Si exclusively as a scaffold element is almost certainly impossible.
−Despite the potentially rich chemistry of silicon, direct substitution of C for Si in organic molecules is often impossible (Section 3.1; Section 3.2).−Formation of many biologically crucial functional groups is much less favorable for silicon than for their carbon counterparts (e.g., unsaturated silicon structures are generally only stable at cryogenic temperatures, if at all) (Section 3.2.1).−Due to the very high affinity of silicon to oxygen the most common stable polymers of silicon are built from a meshwork of Si–O chains instead of linear Si–Si silane polymers (Section 3.1).−The excess cosmic abundance of elemental oxygen ensures that the great majority of the available silicon is almost exclusively, and stably, bonded to oxygen (in the form of unreactive silica). Therefore, while “carbon chemistry is the chemistry of life, silicon chemistry is the chemistry of rocks” (Section 4.1). The substantial energy needed to turn rocks into life, compared to that needed to turn CO_2_ into life, argues against silicon.The vast potential theoretical space of silicon chemistry is almost entirely unstable in water, and hence not available to a biochemistry based on water as a solvent (Section 3.2.1; Section 4.1).Earth’s life silicon biochemistry is extremely chemically limited. In all of Earth’s life, the silicon atom is bonded exclusively to oxygen, forming a Si–O single bond. No naturally occurring life forms synthesize bonds between silicon and any other atom (Section 4.1).The usage of Si in the capacity of a rare heteroatom for water-based biochemistry can, in principle, be possible. The water stability of Si–C and Si–O bonds makes silicon attractive as a carrier of specialized biological functions. We postulate that some small number of organosilicon natural products with rare silicon heteroatoms await discovery (Section 4.1).The energies of formation of Si-containing compounds are generally much higher, and, therefore, less favorable, than their carbon counterparts. The thermodynamics puts additional, although not absolute, constraints on the potential of silicon-based life (Section 3.3).Any sort of biochemistry is implausible in cryogenic solvents, because of solubility limits (Section 4.4).Going forward, we should think about silicon as a contributor to biochemistry (as a common heteroatom in sulfuric acid and a rare heteroatom in water solvent) rather than a main building block of life.

During our review, we have synthesized one new key finding.

A larger fraction of the silicon chemical space is stable in concentrated sulfuric acid than in water. Such greater diversity of possible stable silicon molecules could be exploited by hypothetical sulfuric-acid-based life (e.g., a hypothetical, strictly aerial biosphere living in the sulfuric acid clouds of Venus). Even though carbon would still dominate, silicon could be widely used as a heteroatom component in sulfuric acid biochemistry (Section 4.3).

Even if the most serious objections to the biological use of silicon as a heteroatom can in principle be circumvented, the question of sufficient evolutionary pressure for the use of silicon over carbon as a main component for life still remains.

## Figures and Tables

**Figure 1 life-10-00084-f001:**
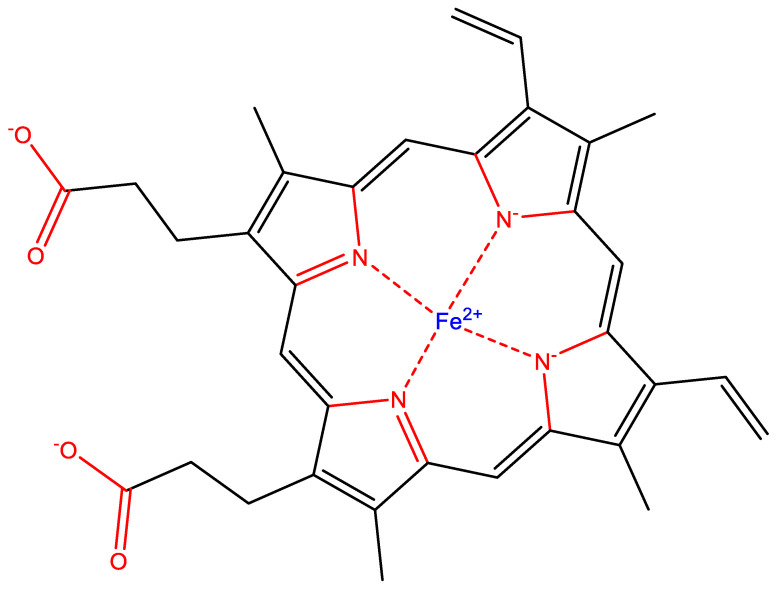
Scaffolding elements and heteroatoms build biochemicals. Scaffolding elements are responsible for creating the overall structure and shape of molecules, while heteroatoms enable necessary chemical diversity, reactivity and directional bonding. For example, in heme (a crucial molecule used by virtually all life on Earth), carbon is a scaffolding element and O and N elements are heteroatoms that allow for the necessary reactivity and directional bonding required for coordination of catalytically important iron ion (Fe^2+^). Heteroatoms and bonds to them are colored red, scaffold elements (carbon) and bonds are colored black.

**Figure 2 life-10-00084-f002:**
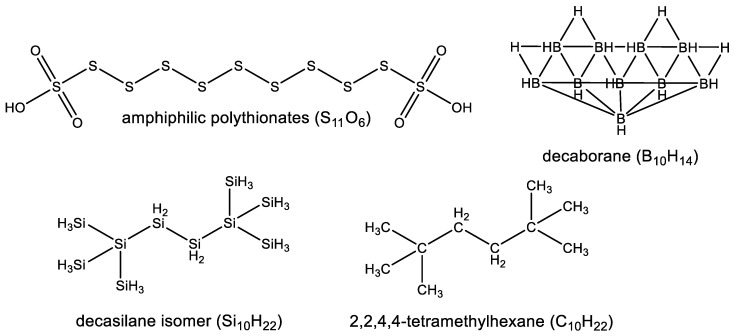
Comparison of four (sulfur, boron, silicon and carbon) scaffolding elements and their pros and cons as main building blocks of biochemistry. Sulfur (e.g., amphiphilic polythionates [11]) forms chains with itself, and in an alternation with carbon, nitrogen, or oxygen (heteroatoms), but has a very limited branched structure, which severely limits the diversity of possible shapes in sulfur-based molecules. Boron (e.g., decaborane [12]) forms chains with itself, and in an alternation with carbon, nitrogen, or oxygen (heteroatoms), but forms clusters of atoms rather than smaller isolated molecules (a problem which is opposite to sulfur). Carbon (e.g., decane isomer) and silicon (e.g., decasilane isomer [15]), on the other hand, form chains with themselves, form chains in alternation with various heteroatoms, and can form diverse linear or branched structures. Out of the possible alternative scaffolding elements, silicon appears to be the most promising choice as a substituent for carbon in biochemistry.

**Figure 3 life-10-00084-f003:**
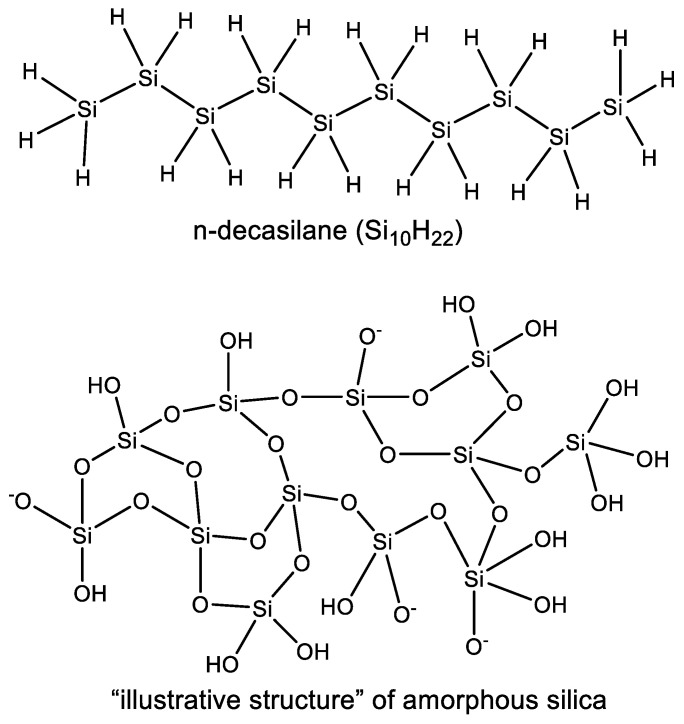
Polymerization of silicon in oxygen-rich aqueous environments leads to a meshwork of Si–O chains (e.g., “illustrative structure” of amorphous silica) and not linear polymers like silanes (e.g., n-decasilane (Si_10_H_22_)). As a result, Si chemistry in oxygen-rich environments (e.g., water) ultimately leads to silica (SiO_2_), a refractory solid.

**Figure 4 life-10-00084-f004:**
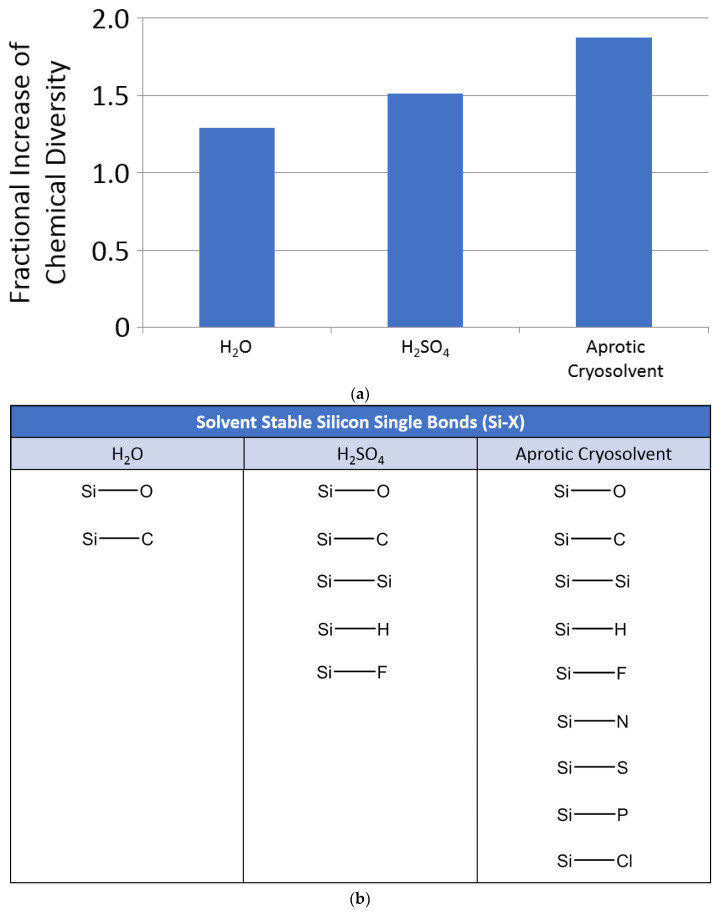
Theoretical chemical diversity of silicon chemistry. (**a**) Relative size of the chemical space with silicon included in SPONCH + F + Cl chemistry, compared to the chemical space without silicon. Y axis: size of the chemical space of chemical structures with up to six non-hydrogen atoms predicted to be stable in respective solvents generated by including silicon chemistry. The relative size of the chemical space is calculated as S/N, where S is the number of molecules in the chemical space with silicon and N is the number without silicon. See main text and Appendix E.1 for details. (**b**) Silicon single bonds generally stable in H_2_O, H_2_SO_4_ and aprotic cryosolvent (a qualitative assessment, based on the chemical reactivity of silicon compounds discussed in detail in Section 4). Very few Si-containing chemical bonds are stable in water. Note that, while some Si–O bonds are generally stable to hydrolysis in water (e.g., in the context of C–Si bonds, like C–Si–OH), many Si–O-bond-containing functional groups are not (e.g., O–Si–O–C), further limiting the scope of Si chemistry available for life in water. Sulfuric acid can support larger number of Si-containing bonds, and, as a consequence, more diverse silicon chemistry. Virtually all chemistry is stable in aprotic cryosolvents.

**Figure 5 life-10-00084-f005:**
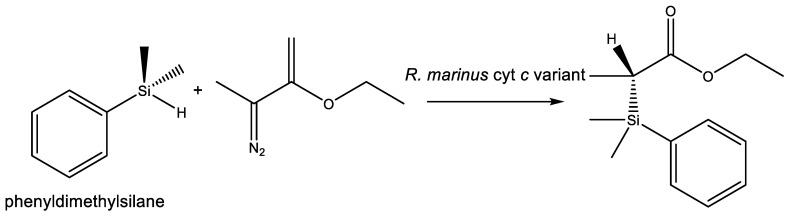
Formation of chiral Si–C bond catalyzed by a laboratory-evolved variant of *Rhodothermus marinus* cytochrome *c* [83,84].

**Figure 6 life-10-00084-f006:**
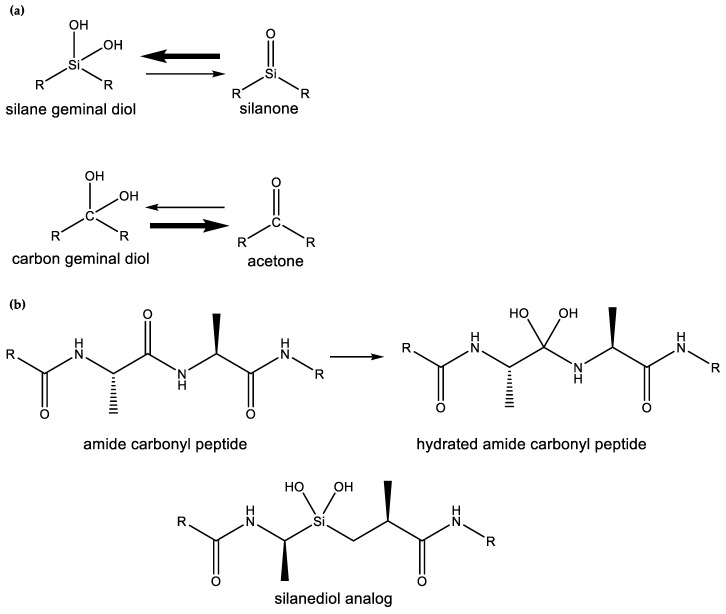
Different bonding preferences between carbon and silicon. (**a**) Silanones (double-bonded Si=O functional groups), are not favored as compared to geminal silicon diols, which, in contrast to carbon germinal diols, are stable. (**b**) Geminal silicon diols mimic the unstable hydrated carbonyl. Silanediol analogues do not have stable carbon analogues and have potent biological functions, e.g., as protease inhibitors, which could be utilized by water-based life. Figure modified from [99].

**Figure 7 life-10-00084-f007:**
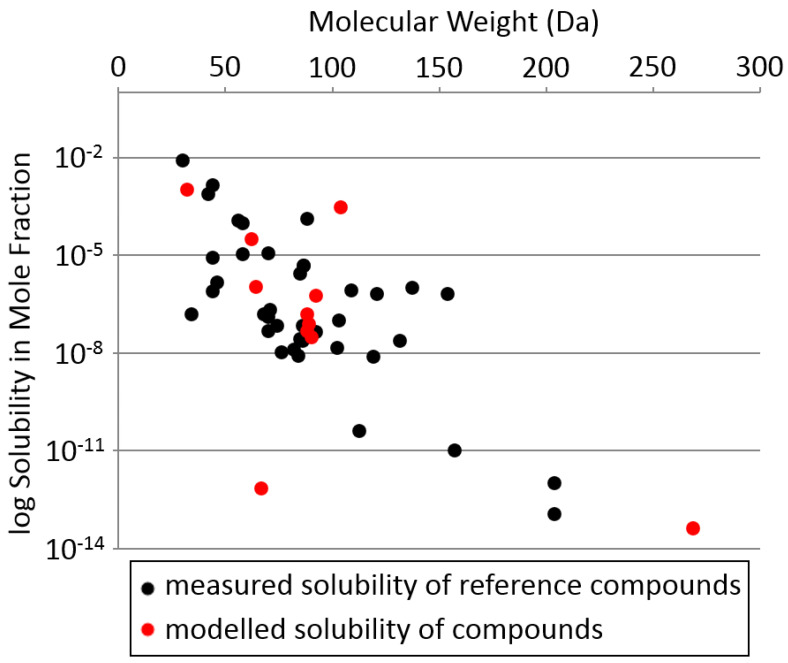
Solubility of compounds vs. molecular weight in liquid nitrogen. Y axis: solubility in mole fraction plotted on log scale. X axis: molecular weight in daltons (Da). Black circles: solubility of measured reference compounds. Red circles: modeled solubility of molecules in Table 2. The results are an “order-of-magnitude” indication of solubility, not a quantitative prediction. In all cases (measured and modeled), the solubility of molecules in liquid N_2_ is very low and decreases gradually with increasing molecular weight. Solubility data for the reference compounds: [120,122,123,124,125,126,127,128,129,130,131,132,133,134,135,136,137,138,139,140,141,142].

**Figure 8 life-10-00084-f008:**
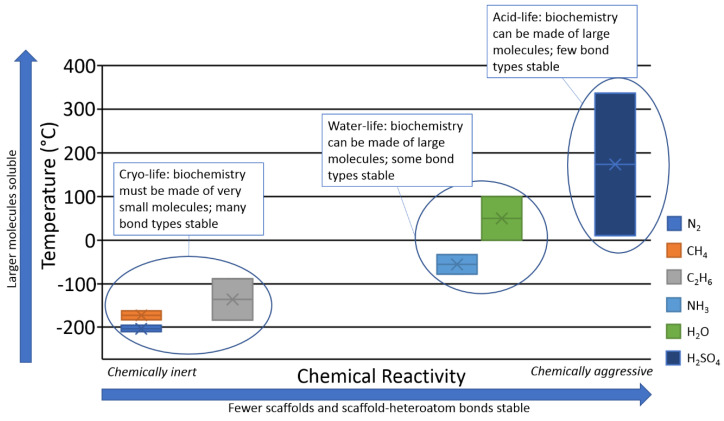
Schematic representation of the viability of different solvents for complex biochemistry, as a function of chemical reactivity. Y axis: Temperature in °C. X axis: Qualitative estimation of chemical reactivity of a solvent. The height of each bar represents the temperature range of a liquid phase of a solvent (at 1 bar).

**Table 1 life-10-00084-t001:** An overview of the basic physico-chemical properties of carbon and silicon, including their respective bond and common functional group reactivity. Data based on values collected in [26,28,29]. *SMILES description of multiple bonds is used throughout the paper [30]. For example, C=C is a double carbon–carbon bond; C:C is an aromatic carbon–carbon system; C#C is a triple carbon–carbon bond. For more details on the chemistry of organosilicon compounds and examples of some exceptionally exotic chemistry that silicon atoms can participate in, see Appendix C.

Feature	Carbon	Silicon
**Electronic Configuration**	C=1s^2^ 2s^2^ 2p^2^	Si=1s^2^ 2s^2^ 2p^6^ 3s^2^ 3p^2^
**Covalent Radius**	77 pm	117 pm
**Coordination Numbers for Stable Compounds**	1, 2, 3, 4	3, 4, 5, 6
**Pauling Scale Electronegativity**	2.50 [C^δ-^←H^δ+^]	1.8 [Si^δ+^→H^δ-^]
**Bond Energies (*D*) and Lengths of Biologically Important Bonds and Their Si Equivalents**	C–C: 346 kJ/mol; 154 pm	Si–Si: 222 kJ/mol; 233 pm
	C–O: 358 kJ/mol; 143 pm	Si–O: 452 kJ/mol; 163 pm
	C–N: 305 kJ/mol; 147 pm	Si–N: 355 kJ/mol;
	C–S: 272 kJ/mol; 182 pm	Si–S: 293 kJ/mol; 200 pm
	C–H: 411 kJ/mol; 109 pm	Si–H: 318 kJ/mol; 148 pm
	C–Si: 318 kJ/mol; 185 pm (longer and slightly weaker than C–C)	
**Bonding Geometry**	e.g., N(CH_3_)_3_; pyramidal	e.g., N(SiH_3_)_3_; planar
**Lipophilicity**	Log P of PhCMe_3_: 4.0	Log P of PhSiMe_3_: 4.7
**Chemical Properties of Selected Functional Groups**	C–H; stable in aqueous solution	Si–H; susceptible to hydrolysis, especially under basic conditions (rate strongly dependent on substituents on Si and pH)
	Si–C; stable in aqueous solution; useful pharmacological properties	
	C–O–C; stable in aqueous solution	Si–O–C; susceptible to hydrolysis (rate strongly dependent on substituents on Si and pH)
	C–OH; stable towards condensation, lower acidity	Si–OH; stable, but liable to condensation (rate strongly dependent on substituents on Si and pH), higher acidity
	C–N; stable in aqueous solution	Si–N; susceptible to hydrolysis, especially under acidic conditions (rate strongly dependent on substituents on Si and pH)
	C–S; stable in aqueous solution	Si–S; susceptible to hydrolysis
**Multiple Bonds ***	C=C, C#C, C:C; stable in aqueous solution	Si=Si, Si#Si; few examples of Si=Si and Si#Si known, highly reactive and unstable in aqueous solution; Si:Si; only one hexasilicone system with dismutational aromaticity is known. Remarkably it is relatively air- and thermostable (<200 °C).
	Si=C and Si#C bonds are known, although Si=C bonds are very unstable under almost all conditions; very few Si:C silabenzene systems are stable	
**Penta- and Hexacoordinate Systems**	Unstable	Well known and stable (in some cases even in aqueous solution) as charged and uncharged species

**Table 2 life-10-00084-t002:** Energy of formation of silicon-containing compounds. The energies of formation of Si-containing compounds are generally much higher, and therefore less favorable, than their carbon counterparts. Thermodynamic values were collected from the literature [72,73], including NIST-JANAFF thermochemical tables [74] and calculated by the GAMESS model for entropy (v1.0) [75,76,77]. All calculations are done for molecules in their standard state. XO_2_ corresponds to carbon- or silicon-containing substrates, carbon dioxide and silica, respectively (see Appendix E.2 for details on the modeled chemical reactions).

Silicon Compound	Carbon Analogue	ΔG° Formation kJ/mol (298 K)		ΔG Formation from XO_2_, H_2_O, N_2_, HCl, HF, H_3_PO_4_ + H_2_ (Standard State)	
		Silicon	Carbon	Silicon	Carbon
**Silane** **SiH_4_**	Methane CH_4_	57.2	−51.12	434.17	−141.62
**Disilane** **Si_2_H_6_**	Ethane C_2_H_6_	127.07	−32.25	880.99	−213.23
**Trisilane** **Si_3_H_8_**	Propane C_3_H_8_	185.18	−23.84	1316.06	−295.32
**Tetramethylsilane** **SiC_4_H_10_**	Neopentane C_5_H_12_	−96.13	−22.35	−81.14	−474.8
**Diethylsilane** **SiC_4_H_10_**	n-pentane C_5_H_12_	−46.96	−8.24	−31.96	−460.69
**Trimethylsilanol** **(CH_3_)_3_SiOH**	Tert-butanol (CH_3_)_3_COH	−372.64	−175.46	−28.81	−299.08
**Chlorosilane** **SiH_3_Cl**	Chloromethane CH_3_Cl	−116.104	−59.88	−356.08	−55.15
**Hexachlorodisilane** **Si_2_Cl_6_**	Hexachlorethane C_2_Cl_6_	−970.08	−57.61	355.16	332.73
**Silicon tetrafluoride** **SiF_4_**	Carbon tetrafluoride CF_4_	−1572.71	−888.51	−444.42	−227.67
**Trimethylsilanamine** **(CH_3_)_3_SiNH_2_**	Tert-butylamine (CH_3_)_3_CNH_2_	−146.59	28.49	−41.10	−333.47
**Silylphosphane** **PSiH_5_**	Methylphosphane CH_5_P	22.76	−8.29	604.18	94.68

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
