# Peer review of "On the Potential of Silicon as a Building Block for Life"

_life, 2020, doi:10.3390/life10060084_

Round 1

Reviewer 1 Report

The manuscript by Petkowski et al. is a very well written paper on an important topic on which there is little literature. It is a very thorough study that advances the field and I look very much forward to see it published. There is a definite need for such a paper to which one can point when the possibility of silicon-based life comes up. The manuscript is mostly a review paper, but also includes new material – such as the surprising result that there may be an enhanced role for Si for life in sulfuric acid – possibly quite relevant to the point whether the lower atmosphere of Venus may harbor life as suggested by several authors this paper cites.

I have only a few points which I bring up with the intention to make the paper even better and more comprehensive:

  • The description of Si and its chemistry is excellent, but in various places there could be an improved link to a suitable environment or environmental conditions. I understand that the title says Earth and exoplanets, nevertheless a link to Io (where is thought to exist liquid hydrogen sulfide and many other sulfur species, line 597) or Titan (hydrocarbon solvent methane/ethane, e.g. line 703-713) could be added/improved – similar exoplanets surely exist as well. By the way, the title could be simplified to “On the Potential of Silicon as a Building Block for Life”, but that´s the authors´ call.
  • The paper is usually well-referenced, but the most exhaustive discussion of the possibility of silicon-life that I´m aware of is not included, and this is the book by Schulze-Makuch and Irwin on Life in the Universe: Expectations and Constraints (now in the 3rd edition, 2018), which has nearly a whole chapter of possible silicon-based chemistry and life. The 2006 paper from the same authors that is referenced (Naturwissenschaften) has some of that content included, but only a small fraction.
  • The solvent discussion makes all sense and is excellent, but I´m missing one. In Schulze-Makuch and Irwin (2018, see above) it is mentioned that if Si life would exist, it would likely go together with a hydrocarbon solvent. For Titan this would be a cryo-solvent such as methane and ethane and this has been addressed. But how about methanol as a solvent? It is a polar hydrocarbon and it has a high temperature range. And it can be pretty plentiful. One paper suggested even oceans of methanol existing on early Mars (Tang et al. 2006, Icarus, Early Mars may have had a methanol ocean). I don´t personally give that study much credibility, however, the point is still that methanol can be pretty plentiful. So, to be comprehensive it would be great to see the potential of methanol addressed as a possible solvent for Si, at least in form of a paragraph. Or, if it is not suitable, then explain why not.
  • In the abstract the authors claim that they provide a “comprehensive assessment of the possibility of silicon-based chemistry and they do not shy away from being speculative (which is a good thing!), but if so what about life based on a silicone network or even as silicates – like the (in)famous lavobes and magmobes suggested by Feinberg and Shapiro (Life Beyond Earth, 1980). Of course, it would be a way-out stretch, but in order to be comprehensive this should be mentioned - and we certainly have to make us free from Earth pressures and temperatures when we talk about possible silicon-based life.
  • There is an Appendix on Biochemistry of Silicon in Life on Earth, which is important  to the paper (and I was wondering may be if there is a way to move this into the main text of the paper? – but that´s the authors´ call). Here I miss some of the suggestions advanced by Wainwright to be at least briefly noted or discussed (e.g. 1997 – the neglected microbiology of silicon) or some of the many other authors that indicate that silicon is critical to growth in some organisms (e.g. Das et al. 1992, Metabolism of silicon as a probably pathogenic factor….; Yoshino, 1990, Growth accelerating effect of silicon on P. aeruginosa) and that heme proteins can catalyze the formation of organosilicons in a bacterium (Kan et al. 2016, ….Bringing silicon to life, Science 354: 1048-1051)
  • Lastly, Cairns-Smith - and I´m sure the authors are familiar with his work - suggested in several of his works that the first life on our planet may have used silicon compounds, clay minerals, as templates, possibly also related to the origin of the genetic code. Again, that has not to be a big topic, but in order to be comprehensive, it would be good to mention it and briefly note what the authors think of that idea.

Reviewer 2 Report

The article titled “On the potential of silicon as a building block of biochemistry on Earth and on exoplanets”, constitutes the first comprehensive and systematic attempt to answer the question whether life can be based on scaffold elements other than carbon. The article discusses the probable biochemistry of alternative scaffold candidates such as sulfur, boron, and, particularly silicon, in protic and aprotic solvents (at very low temperatures, as well), and compares them against water-based carbon biochemistry, by taking into consideration (a) known experimental data and (b) designing appropriate modelled chemical reactions. The article concludes that a life based primarily around silicon chemistry is not a plausible option in any known exoplanetary environment. The study makes sound assumptions and draws valid conclusions, supported by extensive referencing.
